# Effect of 3-Carene and the Micellar Formulation on *Leishmania (Leishmania) amazonensis*

**DOI:** 10.3390/tropicalmed8060324

**Published:** 2023-06-16

**Authors:** Audrey Rouse Soares Tavares Silva, Amanda Mendonça Barros Costa, Ricardo Scher, Valter Viana Andrade-Neto, Victor Hugo Vitorino Sarmento, Adriana de Jesus Santos, Eduardo Caio Torres-Santos, Sona Jain, Rogéria de Souza Nunes, Rubem Figueiredo Sadok Menna-Barreto, Silvio Santana Dolabella

**Affiliations:** 1Departamento de Farmácia, Universidade Federal de Sergipe, São Cristóvão 49100-000, Sergipe, Brazil; audreytavares2@gmail.com (A.R.S.T.S.); amandambc@hotmail.com (A.M.B.C.); drisantos_7@hotmail.com (A.d.J.S.); rogeria.ufs@gmail.com (R.d.S.N.); 2Departamento de Morfologia, Universidade Federal de Sergipe, São Cristóvão 49100-000, Sergipe, Brazil; rica.scher@gmail.com; 3Laboratório de Bioquímica de Tripanosomatídeos, Instituto Oswaldo Cruz, Fundação Oswaldo Cruz, Rio de Janeiro 21040-900, Brazil; valterfarmaneto@gmail.com (V.V.A.-N.); eduardocaiotorres@yahoo.com.br (E.C.T.-S.); 4Departamento de Química, Universidade Federal de Sergipe, Itabaina 49506-036, Sergipe, Brazil; vhsarmento@gmail.com; 5Programa de Biotecnologia Industrial, Universidade Tiradentes, Aracaju 49032-490, Sergipe, Brazil; sonajain24@yahoo.com; 6Laboratório de Biologia Celular, Instituto Oswaldo Cruz, Fundação Oswaldo Cruz, Rio de Janeiro 21040-900, Brazil

**Keywords:** monoterpene, antileishmanial, triblock copolymers, nanostructure, cytotoxicity

## Abstract

Leishmaniases are neglected tropical diseases caused by obligate intracellular *protozoa* of the genus *Leishmania*. The drugs used in treatment have a high financial cost, a long treatment time, high toxicity, and variable efficacy. 3-Carene (3CR) is a hydrocarbon monoterpene that has shown in vitro activity against some *Leishmania* species; however, it has low water solubility and high volatility. This study aimed to develop Poloxamer 407 micelles capable of delivering 3CR (P407-3CR) to improve antileishmanial activity. The micelles formulated presented nanometric size, medium or low polydispersity, and Newtonian fluid rheological behavior. 3CR and P407-3CR inhibited the growth of *L.* (*L.*) *amazonensis* promastigote with IC_50_/48h of 488.1 ± 3.7 and 419.9 ±1.5 mM, respectively. Transmission electron microscopy analysis showed that 3CR induces multiple nuclei and kinetoplast phenotypes and the formation of numerous cytosolic invaginations. Additionally, the micelles were not cytotoxic to L929 cells or murine peritoneal macrophages, presenting activity on intracellular amastigotes. P407-3CR micelles (IC_50_/72 h = 0.7 ± 0.1 mM) increased the monoterpene activity by at least twice (3CR: IC_50_/72 h >1.5 mM). These results showed that P407 micelles are an effective nanosystem for delivering 3CR and potentiating antileishmanial activity. More studies are needed to evaluate this system as a potential therapeutic option for leishmaniases.

## 1. Introduction

Leishmaniases are neglected infectious diseases caused by protozoa of the genus *Leishmania* and transmitted by blood-sucking phlebotomine sand flies as a vector-borne disease [1]. Such illnesses are associated with climate and environmental changes, poverty, poor nutrition, poor housing conditions, and the vulnerability of the host’s immune system. It is estimated that 700,000 to 1,000,000 new cases occur annually [2].

Some species of *Leishmania* have tropism for organs, especially the liver, spleen, and bone marrow, causing visceral leishmaniasis, a severe form of the disease that usually leads to the patient’s death if not treated [3]. However, even in treated patients, the average lethality is around 8% in the Americas [4]. On the other hand, some parasite species have dermal tropism and remain in the cutaneous area at the inoculation site or spread to other integument regions, causing localized cutaneous lesions or even diffused or mucocutaneous lesions [5]. These clinical manifestations in the integument are usually not lethal but can promote deformations and/or scars in large body extensions, and mutilations that directly affect the patient´s quality of life [6].

Despite the various forms of leishmaniases, a limited group of drugs such as pentavalent antimonials, amphotericin B, pentamidine, miltefosine, paromomycin, and some azoles are available for treatment [7]. In general, these drugs have drawbacks that can compromise their efficacy, involving severe side effects, discomfort due to the intramuscular or intravenous route of administration, an extended treatment time and thus high financial costs, variable anti-leishmanial effects, and the imminent risk of the emergence of new resistant strains [8].

The limitations related to the pharmacotherapy of leishmaniases and the absence of other forms of control reinforce the urgency for more effective, safe, and low-cost alternative drugs. Natural products stand out in this context, representing approximately 25% of all drugs currently marketed [9,10,11]. Monoterpenes are secondary metabolites with multiple pharmacological properties [12,13], which have demonstrated in vitro and in vivo activities against different species of *Leishmania* [14,15,16,17,18].

3-Carene (3CR) is a monoterpene identified in several vegetal species [19], and it is among the main components of pine tree (*Pinus* genus) and spruce (*Piceaabis*) oil-resin, natural in temperate regions, and pepper (*Piper* genus), common in tropical and subtropical areas [20,21]. In previous studies, the leishmanicidal activity of 3CR has been assessed in vitro against *Leishmania* (*L.*) *donovani* (IC_50_/72 h = 27.0 µg/mL) and *Leishmania amazonensis* (IC_50_/24 h = 72.5 µg/mL) promastigotes [22,23]. However, similar to other monoterpenes, 3CR has high hydrophobicity and volatility, rapid metabolism, and consequent rapid elimination in humans, compromising its use as a therapeutic agent [24]. In this scenario, the use of nanotechnological strategies as drug delivery systems is crucial toovercome these limitations.

Poloxamers are triblock copolymers that, in an aqueous medium, can self-organize into nanometric-sized micelles. Micellar systems (micelles, hydrogels, andorganogels) of poloxamer have been used to solve problems such as low solubility and reduced absorption of drugs; additionally, they offer protection against drug degradation, modified release, and improved efficacy [25]. Furthermore, poloxamer micelles have the most reduced dimensions (10 to 200 nm) among the nanosystems used for drug delivery, which favors circulation in the bloodstream, diffusion in distinct tissues, and cell uptake [26,27,28,29]. Therefore, it is particularly relevant for formulations for leishmaniases, as the drugs must reach the parasite in the parasitophorous vacuole within the host macrophages. 

Thus, this work aimed to develop and characterize micelles containing 3CR to improve antileishmanial activity while also assessing its primary targets in *L. amazonensis* promastigotes by electron microscopy.

## 2. Materials and Methods

### 2.1. Chemical and Reagents

Poloxamer 407 (P407), 3CR (3,7,7-Trimethyl-bicyclo[4,1,0]hept-3-ene,96.1% pure), amphotericin B, resazurin, MTT salt (3-[4,5-dimethylthiazol-2-yl]-2,5-diphenyltetrazolium bromide), the mediums Schneider’s insect, DMEM, and RPMI were purchased from Sigma-Aldrich (St. Louis, MO, USA). All chemicals and solvents were of analytical grade.

### 2.2. Micelle Preparation

Micelles were prepared by dispersing appropriate amounts of P407 (5.15% *w*/*v*) in sterile phosphate-buffered saline (PBS, pH 7.4) under mechanical stirring with a magnetic bar (100 rpm) at room temperature [30]. After one hour of stirring and complete dissolution of the polymer, 3CR was added at a final concentration of 7.3 mM (1 mg/mL) and stirring was continued for another hour. Micelles without the addition of 3CR were also formulated following the same protocol. The formulations obtained (P407 and P407-3CR micelles) were stored and protected from light in an amber bottle and refrigerated at 8 °C. The maximum capacity of P407 micelles (5%) to solubilize 3CR defined the concentration used in the micelles. PBS as a dispersant medium was considered due to the compatibility of this solution with the physiological medium.

### 2.3. Dynamic Light Scattering (DLS) Analysis

The micelles’ hydrodynamic diameter (HD) and polydispersity index (PDI) were determined by dynamic light scattering (DLS, Dynamic Light Scattering) using the Zetasizer Nano ZS equipment (Malvern Instruments^®^, Malvern, UK), with a fixed detector at an angle of 173°. Measurements were performed in a quartz cuvette using undiluted samples at 25 and 37 °C [31].

### 2.4. Rheological Analysis

The rheological characterization of micelles was carried out in a Discovery Hybrid HR-1 rheometer (TA Instruments, New Castle, DE, USA). The flow test was performed using a cone-plate geometry of 60 mm diameter, and a 0.050 mm distance between the clone and the plate, with a 2° angle and a running time of 4 min for each sample. The shear rate ranged from 0 to 200 s^−1^, keeping the temperature controlled at 25 °C. Each sample was carefully applied to the bottom plate of the rheometer, ensuring minimal shear and allowing a rest time (relaxation of the induced tension before analysis) of one minute before each determination. The data obtained were analyzed using Origin Pro Software v.8 [31].

### 2.5. Promastigotes Cultivation and Host Cell Cultures Maintenance

*L.* (*L.*) *amazonensis* promastigotes (strain MHOM/BR/77/LTB0016) were cultivated in Schneider’s insect medium (Gibco^®^, Thermo Fisher Scientific, Waltham, MA, USA) supplemented with 10% fetal bovine serum (FBS), 100 µg/mL of streptomycin, and 100 U/mL of penicillin maintained at 26 °C in a BOD (Biochemical oxygen demand) until the sixth passage.

L929 fibroblasts were cultured in DMEM medium (pH 7.4) supplemented with 10% inactivated FBS and 1% antibiotic solution (5.000U/mL penicillin and5 mg/mL streptomycin) maintained in an atmosphere of 5% CO_2_ at 37 °C and 95% humidity.

### 2.6. Antipromastigote Activity

*L.* (*L.*) *amazonensis* promastigotes (1 × 10^6^ cells/mL) in logarithmic phase of growth were distributed in 96-well plates and treated with 3CR or micelles (0.1–3.7 mM) for 48 h at 26 °C. Promastigotes incubated in the absence of the formulation were used as a negative control, and amphotericin B (0.03–1 µM) was used as a positive control. The viable parasites were assessed by resazurin fluorescent assay [32]. After the treatment, resazurin was added to each well at a final concentration of 50 µM, and the plates were again incubated for 2 h at 26 °C. The fluorescence was measured using a spectrophotometer (Biotek, Winooski, VT, USA, model Synergy H1) at 560 nm excitation and 590 nm emission. The concentration that inhibits 50% of the parasites (IC_50_) was obtained by nonlinear regression of the concentration–response curve using GraphPad Prism 5.0 software. The experiments were performed in triplicates in at least three independent biological experiments.

### 2.7. Antiamastigote Activity

Resident macrophages from BALB/c mice were obtained by peritoneal washing with 5 mL of cold RPMI medium. The cell suspension was adjusted to 2 × 10^6^ cells/mL (0.4 mL/well), plated in Lab-Tek eight-chamber slides (Nunc, Roskilde, Denmark), and incubated at 37 °C with 5% CO_2_ for 1 h. Stationary-phase *L.* (*L.*) *amazonensis* promastigotes were added at parasite/macrophage ratio of 3:1 for 3 h. Later, the wells were washed three times to remove non-interiorized parasites and then incubated under the same conditions. After 24 h, the infected macrophages were treated with 3CR, micelles (0.02–2.9 mM), and amphotericin B (0.03–1 µM) for 72 h. Next, the slides were stained using rapidpanoptic stain (Newprov, Curitiba, Brazil), and the infection rate was determined by counting under optical light microscope [32]. The experiments were performed in three biological replicates, with two technical replicates, and the number of amastigotes was determined by counting at least 100 macrophages in each one. The results were expressed as an infection index (% infected macrophage × number of amastigotes/total number of macrophages). The IC_50_/72 h was determined by nonlinear regression analysis in the software GraphPad Prism 5 [33].

### 2.8. Cytotoxicity to the Host Cells

Cytotoxicity of 3CR and micelles was evaluated on two distinct mammalian cells. In a first set of experiments, MTT method with modifications was employed [33]. L929 fibroblasts (2 × 10^4^ cells/well) were added to 96-well plates and kept in a 5% CO_2_ atmosphere at 37 °C for 24 h. These cultures were treated with 0.2 to 3.7 mM of 3CR or only micelles and were incubated under the same conditions for 24 h. Then, the cells were washed with PBS, and 200 µL of MTT solution (0.5 mg/mL) was added per well for 3 h. MTT was removed, and the formazan crystals were solubilized with 200 µL DMSO. Optical density was measured at 570 nm.

Alternatively, murine peritoneal macrophages were employed using the resazurin assay [33]. Resident peritoneal macrophages were collected as described before. The cell suspension was adjusted to 2 × 10^6^ cells/mL, distributed in 96-well plates, and treated with 3CR, micellar systems (0.04–5.9 M), and amphotericin B (0.15–20.0 µM) for 72 h at 37 °C in an atmosphere of 5% CO_2_. Later, steps similar to those described for the antipromastigote test were carried out. All CC_50_ values were determined by nonlinear regression of the concentration–response curve using the GraphPad Prism 5.0 program. The experiments were performed in triplicates in at least three independent biological experiments.

In order to compare the biological activity and toxicity to the host, the selectivity index (SI = CC_50_ 72 h/IC_50_ 72 h) was calculated.

### 2.9. Transmission Electron Microscopy Analysis

*L.* (*L.*) *amazonensis* promastigotes (5 × 10^6^ cells/mL) were treated with 488.1 mM 3CR for 48 h at 26 °C. The parasites were fixed with 2.5% glutaraldehyde diluted in 0.1 M sodium cacodylate buffer (pH 7.2) for 40 min at 25 °C, and then post-fixed with a solution of 1% OsO_4_ containing 0.8% potassium ferricyanide and 2.5 mM CaCl_2_ in the same buffer for 20 min at 25 °C. The samples were dehydrated in an ascending acetone series and embedded in PolyBed 812 resin. Ultrathin sections were stained with uranyl acetate and lead citrate and the examination was performed in a Jeol 1200 EX transmission electron microscope (Tokyo, Japan) at National Center for Structural Biology and Bioimaging, Federal University of Rio de Janeiro.

### 2.10. Statistical Analysis

The analysis of variance (ANOVA) was applied to determine the most relevant results. The values obtained were compared and separated by the Bonferroni test. Statistical analysis was conducted using Graph Pad Prisma 5.0. Values with *p* < 0.05 were considered significant.

### 2.11. Ethics Statement

The use of animals and our experimental procedure arein accordance with Brazilian Law 11.794/2008 and regulations of the National Council of Animal Experimentation Control (CONCEA). Mice were housed at a maximum of 4 per cage, kept in a specific-pathogen-free (SPF) room at 20 to 24 °C under a 12 h light and dark cycle, and provided sterilized water and rodent chow ad libitum. All animal procedures were reviewed and approved by Fiocruz Committee of Ethics in Animal Research (L-002/2022), according to resolution 196/96 of the National Health Council of Brazilian Ministry of Health.

## 3. Results

### 3.1. Physical–Chemical Characterization of Micelles

#### 3.1.1. Micelles Obtained and Micelle Size

The micelles obtained were initially evaluated by visual inspection and light microscopy. Both micelles’ formulations were transparent, uniform, and colorless, without particulate matter or phase separation. To determine the size of the micelles, the hydrodynamic diameter (HD) and the polydispersion index (PDI) of the micelles were analyzed. These parameters were evaluated under the influence of temperatures at 25 °C (room temperature) and 37 °C (physiological temperature) both before and after 3CR incorporation [31] (Table 1).

The micelles presented nanometric sizes that varied from 18.48 to 26.81 nm and PDI from 0.206 to 0.463, relative to a size distribution that varied from monomodal to multimodal. In multimodal samples, HD reflects more than 75% of micelles. The incorporation of 3CR caused a significant increase in HD and PDI, while the increase in temperature caused a reduction in HD and PDI in both micelles, with and without 3CR.

#### 3.1.2. Rheological Behavior

The flow test was carried out to observe the rheological behavior of the micelles with and without 3CR (Figure 1). The micelles presented flow curves in which the shear stress and the shear rate present linear dependence. These data classify the systems as Newtonian fluid [34]. From the linear fit of the curves, it was possible to obtain viscosity values around 2.8 mPa.s and 2.5 mPa.s for P407 and P407-3CR, respectively, indicating that the micelles showed low viscosity and the incorporation of 3CR did not change the flow rate.

### 3.2. Bioassays

#### 3.2.1. Leishmanicidal Activity in Promastigotes

The antipromastigote effect was evaluated after 48 h of treatment and expressed as IC_50_/48 h. No statistical differences were observed between P407-3CR micelles and 3CR, indicating that the incorporation of the monoterpene in the micelles maintained the anti-protozoa activity (Figure 2 and Table 2).

#### 3.2.2. Leishmanicidal Activity on Intracellular Amastigotes and Cytotoxicity to Host Cells

The antiamastigote effect was evaluated in infected macrophages after 72 h of treatment. 3CR showed low activity on intracellular amastigotes and it was excluded from the subsequent analysis. Both micelles demonstratedan effect on the parasites; however, P407-3CR was two-fold more active than the pure micelle (Figure 3 and Table 3).

The cytotoxicity of 3CR and micelles was evaluated in L929 cells (Figure 4) and peritoneal macrophages (Table 3). Using L929 cells model, 3CR was potentially toxic at a concentration above 1.8 mM. However, the P407 and P407-3CR micelles were not cytotoxic, showing viability above 80%, even at the highest concentrations of 3CR.

In primary macrophages, 3CR or micellar systems were also not cytotoxic, even with a longer treatment time (72 h), with CC_50_/72 h values greater than 2.6 mM (Table 3). As can also be observed in Table 3, P407-3CR showed a higher SI value than the micelle alone, being 5.1-fold more active in the parasite than in the macrophage. The control amphotericin B presented the highest SI value (>66.6).

#### 3.2.3. Ultrastructural Analysis

The transmission electron microscopy technique was employed to assess the possible target organelles of 3CR. Considering that 3CR and 3CR-loaded micelles showed similar antipromastigote activities, the ultrastructural evaluation was only performed on 3CR. The evaluation of untreated promastigotes showed typical morphological features, including the elongated body and classical aspects of organelles such as the mitochondrion, kinetoplast, nucleus, and flagella (Figure 5A,B). The treatment with an IC_50_/48 h dose of 3CR led to bizarre multiple nuclei and kinetoplast phenotypes, showing an even number of these organelles (usually four per cell) and the formation of innumerous cytosolic elongated invaginations (Figure 6A–F), all suggestive of the impairment of parasite mitosis. The presence of lipid droplets was also easily detected after the treatment with 3CR (Figure 6C–E). Despite the high number of kinetoplasts, thekDNA network was wellpreserved after the treatment (Figure 6C–F).

## 4. Discussion

Natural products derived from plants, animals, or microorganisms have been playing a relevant role for biotechnological purposes, including in the development of alternative therapeutical strategies, either as a starting point for synthetic modifications or as drugs [11,35]. Aiming to offertherapeutic viability and improve the antileishmanial activity of 3CR, poloxamer P407 micelles loaded with 3CR were developed. This poloxamer has been tested on old drugs (propranolol, lidocaine, and amphotericin B) and new compounds with potential therapeutic (8HQ and clioquinol) [36,37,38,39]. P407 at a concentration of 5.15% was sufficient to form micelles capable of incorporating 7.3 mM of 3CR, resulting in a homogeneous dispersion of the monoterpene in water, similar to that observed by Lucia et al. [40].

The formed micellar structure was initially verified by DLS analysis, showing nanometric aggregates compatible with the size of poloxamer micelles. The micelles were sensitive to temperature, showing smaller sizes at 37 °C. The micelles formulated present polydisperse systems with median or low PDI [41]. According to WU et al. [42], only PDI above 0.5 indicates a wide distribution. The increase in temperature also resulted in a lower PDI. Only the micelles without 3CR at 37 °C presented a monodisperse system. The reduction in the mean diameter and polydispersity of the micelles observed with increasing temperature is related to the dehydration of the polypropylene oxide (PPO) units and the consequent reduction in volume. Similar results were described by Akkari et al. [43], who also formulated a micellar system with P407 and observed lower HD and PDI at a higher temperature.

The incorporation of 3CR into micelles caused an increase in HD and PDI. By chemical affinity, the 3CR must have been accommodated in the micellar nucleus, formed by the poloxamer’s hydrophobic portions (PPO). Hydrophobic compounds with log P above 3.5 (3CR log P 4.38) remain completely solubilized in micelles dispersed in an aqueous medium [44,45]. The incorporation of the compound possibly led to core swelling and an increase in the number of unimers per micelle to compensate for the larger core volume, resulting in a higher aggregation number. Grillo et al. [44] observed an increase in the nucleus radius and aggregation number of P407 micelles after monoterpene linalool was incorporated. Akkari et al. [31] also observed growing HD and PDI after ropivacaine loading into P407 micelles.

An efficient antileishmanial therapy requires the drugs to diffuse widely to eliminate the parasite via the host’s immune system or directly reach the parasite in the infected host’s cells. Therefore, particle size is especially relevant for this kind of approach. Particles ranging from 10 to 70 nm often have good tissue distribution because they spend more time in the bloodstream. Those smaller than 100 nm can be internalized by endocytic activity [27,46]. The size of the micelles developed in this study (18.48 to 26.81 nm) is within the expected range, aiming to keep them in the bloodstream for a longer time, thus resulting in better tissue diffusionand uptake to reach the intracellular parasite.

Micellar dispersions, with and without 3CR, showed Newtonian fluid behavior with low viscosity, as expected. Some studies have shown that Newtonian fluid behavior can be observed even at high concentrations of poloxamers, depending on the temperature [47,48]. Pharmaceutical formulations that behave like Newtonian fluid have broad applicability in terms of route of administration, favoring the use of oral, parenteral, and topical routes conventionally used to administer leishmanicidal drugs.

*L.* (*L.*) *amazonensis* is among the main species that cause cutaneous leishmaniasis in the Americas. It is related to the severe clinical form of diffuse cutaneous leishmaniasis. Furthermore, this parasite species has already been isolated from dogs and humans with visceral leishmaniasis, a clinical manifestation historically associated with *L.* (*L.*) *infantum* [49,50], highlighting its high pathogenic capacity. In previous studies, 3CR, a monoterpene hydrocarbon, showed superior activity in *L.* (*L.*) *amazonensis* compared to several other secondary metabolites that presented more reactive molecular structures, excluding compounds with a phenolic moiety [22]. Furthermore, 3CR is a limonene molecular isomer that shows antileishmanial action [18], justifying its choice for this study. In promastigotes, micelles containing 3CR were the most active in comparison with pure 3CR or empty micelles, presenting the lowest IC_50_ values. The micelles without the monoterpene also reduced the viability of the parasite, but not enough to determine its IC_50_. The biological response modification of poloxamers has already been documented for some *Leishmania* species [28].

Similarly, against the intracellular amastigotes, P407-3CR micelles showed better antiparasitic activity with a SI greater than five. Previous reports with P407 poloxamer micelles have also shown promising results concerning antileishmanial activity. Paromomycin encapsulated in P407 micelles showed higher antipromastigote activity on *L.* (*L.*) *infantum* and *L.* (*L.*) *major* when compared to the free drug [47]. On the other hand, previous studies have shown that drugs loaded in poloxamer micelles only reproduce the leishmanicidal activity of the drug alone but reduce the cytotoxicity. *L.* (*L.*)*donovani* amastigotes treated with amphotericin B in suspension and amphotericin B encapsulated in P407 micelles showed similar results with an IC_50_/48 h in the range of 0.1 µg/mL; however, the micellar system presented lower cytotoxicity on J774A.1 macrophages [51]. Likewise, Costa et al. [52] encapsulated the monoterpene carvacrol in a P407 micellar system and did not observe a reduction in the IC_50_/48h value in *L.* (*L.*) *amazonensis* amastigotes, but a significant decrease in cytotoxicity was observed, resulting in a better SI.

The cytotoxicity in this study was evaluated in primary culture macrophages and a fibroblast lineage. Macrophages play a central role in the maintenance, replication, and elimination of *Leishmania* [53], while fibroblasts are present in most tissues and related to tissue integrity and wound healing [54]. P407 and P407-3CR micelles were not cytotoxic to L929 cells at any evaluated concentrations. Furthermore, the cytotoxicity presented by 3CR in the highest concentrations (1.8 and 3.7 mM) was neutralized when incorporated into the micelles. Poloxamer 407 has shown low cytotoxicity, even at much higher concentrations. In the study by Niyompanich et al. [55], L929 cells were treated with P407 (20%) in water for up to seven days and did not demonstrate cytotoxicity. In this study, 3CR or micellar systems were not toxic to peritoneal macrophages, presenting a high value of CC_50_. Oyama et al. [28] also observed similar results when treating BALB/c macrophages with 407 micelles at 48 h. Anyway, the SI values for 3CR in micelles or not were far from ideal, much lower than the SI for amphotericin B.

To evaluate the mechanism of action of 3CR in *L.* (*L.*) *amazonensis*, an analysis was carried out bytransmission electron microscopy in treated promastigotes. Our ultrastructural data pointed to the impairment of the parasite division process induced by 3CR, including multiple nuclei and kinetoplasts. Several cytoplasmic invaginations were also observed, suggesting that the cytokinesis of protozoa was not completed. Promastigotes of *L.* (*L.*) *amazonensis* with induced resistance to vinblastine and wild-type parasites treated with vinblastine also showed multinucleated and multilobed cells, which indicate impaired cytokinesis [56]. Similarly, *Trypanosoma cruzi* treated with vinca alkaloids showed multiple nuclei and kinetoplasts, resulting from a reversible blockage of cytokinesis [57]. Vinblastine is a compound that binds to tubulin, a significant protein in trypanosomatids involved in important events such as nuclear and kinetoplast division and cytokinesis. Taxol, a drug that stabilizes microtubules, also affected cytokinesis and cell cycle progression in G2/M without interfering with the nuclear division of *L.* (*L.*) *donovani* promastigotes [58]. In trypanosomes, cytokinesis is dependent on DNA synthesis but can occur independently of mitosis [59]. Similar mechanisms have been observed in *Leishmania*, but the order and time in which the events occur vary according to the species [60,61].

Compounds that interact with tubulin have been shown to alter the cell cycle in the G2/M phase and inhibit cytokinesis [62]. It is possible that 3CR interacts with tubulin and/or, due to its lipid characteristic, interacts with membrane sphingolipids, causing cytokinesis inhibition. Castro et al. [63] observed promastigotes of *L.* (*Viannia*) *braziliensis* with three or more nuclei and incomplete cytokinesis after treatment with myriocin, an inhibitor of sphingolipid synthesis. Sphingolipids are a class of lipids involved in many biological processes in trypanosomatids and the inhibition of sphingolipid synthesis showed that these lipids are essential for the conclusion of the parasite’s cytokinesis.

Electron microscopy showed an increase in lipid bodies in parasites treated with 3CR. Godinho et al. [64] also observed the accumulation of lipid bodies in the cytoplasm of *L.* (*L.*) *amazonensis* promastigotes after treatment with the synthetic compound alkyl-phosphatidylcholine-dinitrate (TC95), which suggests an alteration in the phospholipid and sterol content. These authors also identified alterations in the cell cycle, with the majority of the cells arrested in the G1 phase and a significant increase in the number of cells in the G2 phase. These aberrant phenotypes were unable to complete cytokinesis and prevent the start of mitosis, as seen through electron microscopy [64].

A pattern of changes similar to our results was observed in *L.* (*L.*) *amazonensis* promastigotes after the treatment with an alkaloid β-carboline (β-CB). The accumulation of lipid bodies, cells with three or more nuclei, and the inhibition of cytokinesis without interfering with the duplication of cell structures were observed. The alkaloid impairs cytokinesis, preventing parasite proliferation through acytostatic effect [60]. Considering our ultrastructural analysis and comparing similar results obtained in other studies [60], it is possible that 3CR performs its antileishmanial effect by cytostatic and not by cytotoxic. However, several other trials are needed to confirm this hypothesis.

## 5. Conclusions

P407-3CR micelles demonstrated activity against both promastigotes and intracellular amastigotes. The activity against promastigotes was similar to that observed by 3CR alone. However, the activity against amastigotes was increased by P407 micelles, suggesting that 3CR reached the parasite within the parasitophorous vacuole. The P407-3CR micelles could also reduce the cytotoxicity in both fibroblast and macrophage cells compared to 3CR alone. The first assessment of its mechanism of action pointed to the interference of the cell division process, as evidenced by our ultrastructural data.

## Figures and Tables

**Figure 1 tropicalmed-08-00324-f001:**
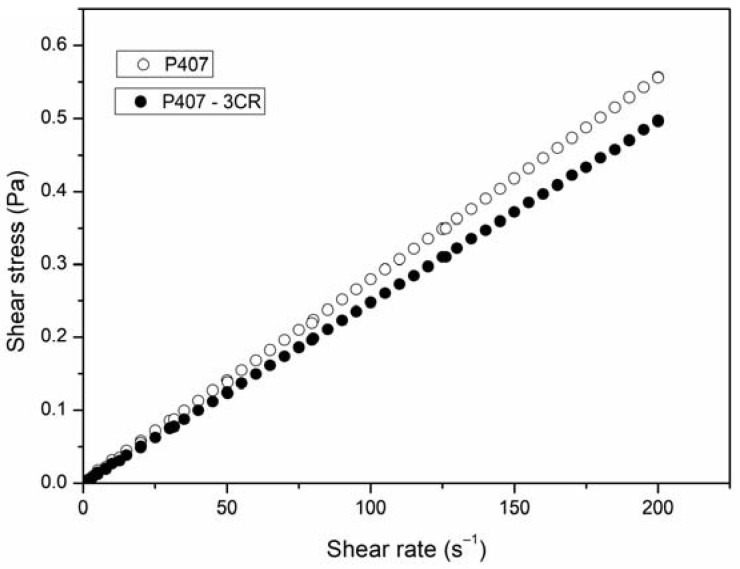
Rheogram for unloaded and 3CR-loaded micelles at 25 °C. Results are representative of three independent experiments. 3CR: 3-Carene.

**Figure 2 tropicalmed-08-00324-f002:**
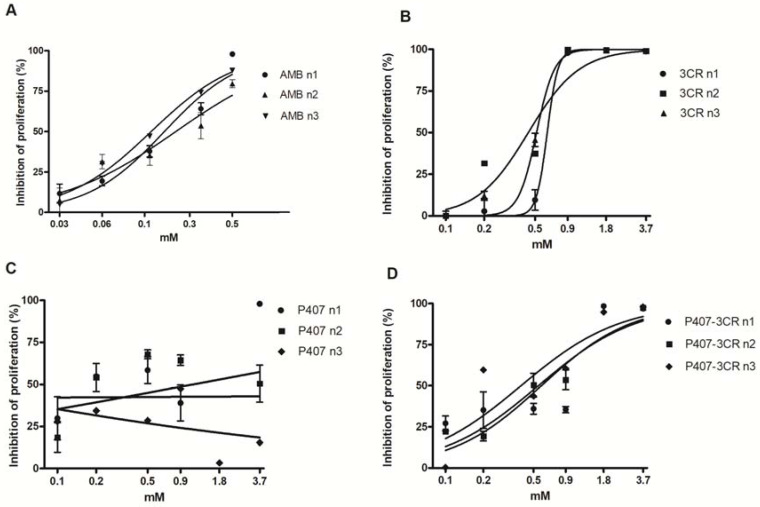
Effect of AMB (**A**), 3CR (**B**), P407-3CR (**D**), and P407 (**C**) micelles on the viability of *Leishmania* (*L.*) *amazonensis* promastigotes. Cells (1 × 10^6^ cells/well) were incubated in 96-well plates for 48 h in the presence of 3CR, P407-3CR, and P407 micelles at different concentrations. Data are expressed as mean ± standard error (*n* = 3). AMB: amphotericin B.

**Figure 3 tropicalmed-08-00324-f003:**
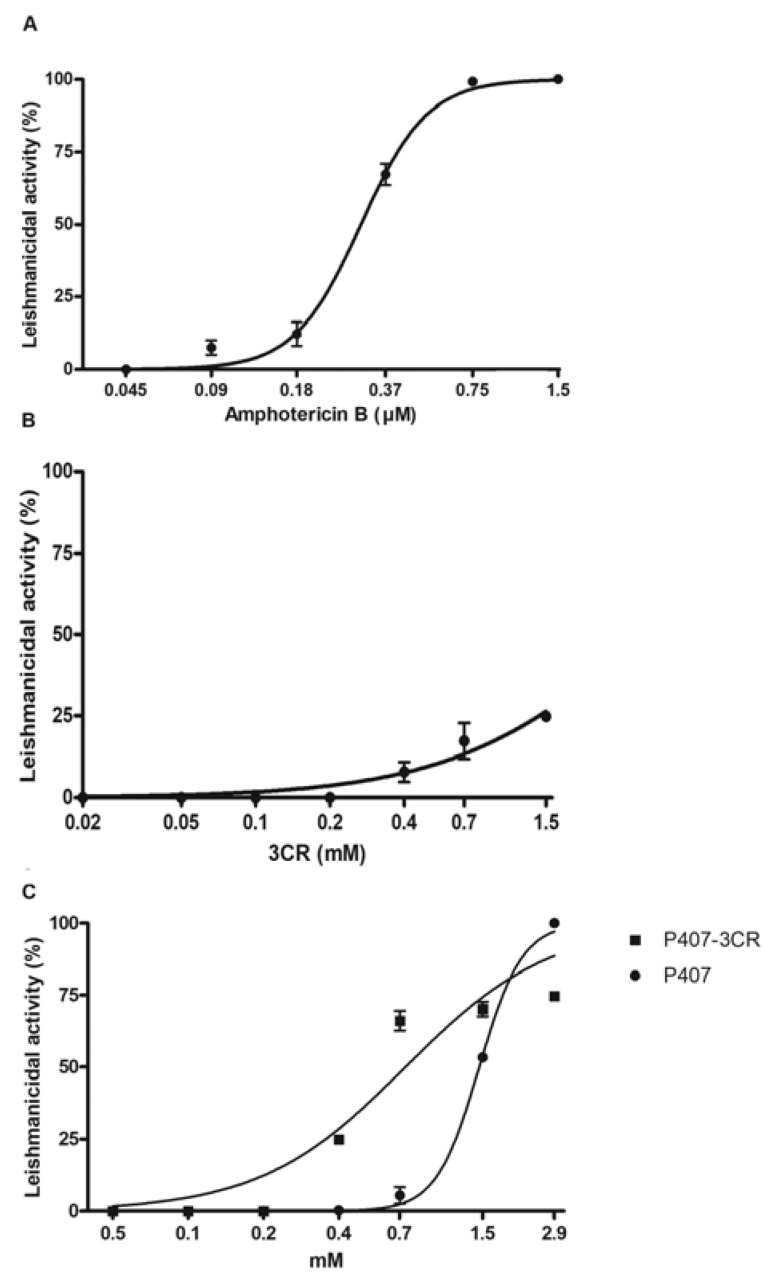
Antiamastigote activity of 3CR and its micelles. Peritoneal macrophages infected with *Leishmania* (*L.*) *amazonensis* were treated with amphotericin B (**A**), 3CR (**B**), P407 (**C**), and P407-3CR (**C**) for 72 h at 37 °C. Data are representative of three independent biological experiments. Data are expressed as mean ± standard error (*n* = 3).

**Figure 4 tropicalmed-08-00324-f004:**
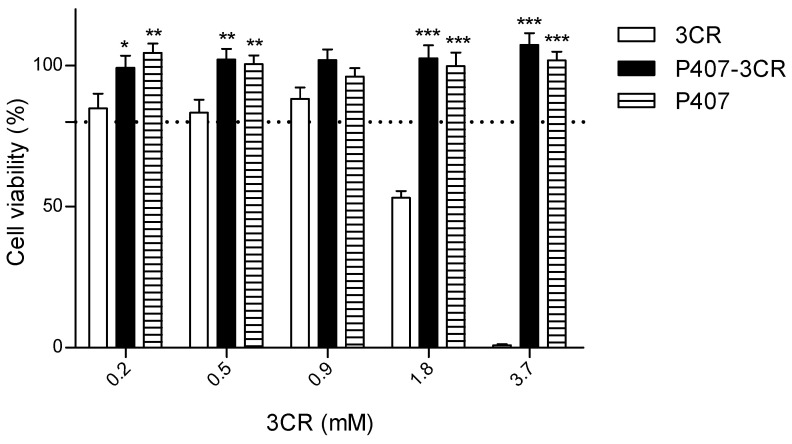
Effect of 3CR, P407-3CR, and P407 micelles on the L929 fibroblasts viability. Cells (2 × 10^4^ cells/well)were incubated in 96-well plates for 24 h in the presence of 3CR, P407-3CR, and P407 micelles at different concentrations. The dashed line represents 80% cell viability. Data are expressed as mean ± standard error (*n* = 3). Statistically different from 3CR * *p* < 0.05, ** *p* < 0.01 and *** *p* < 0.001.

**Figure 5 tropicalmed-08-00324-f005:**
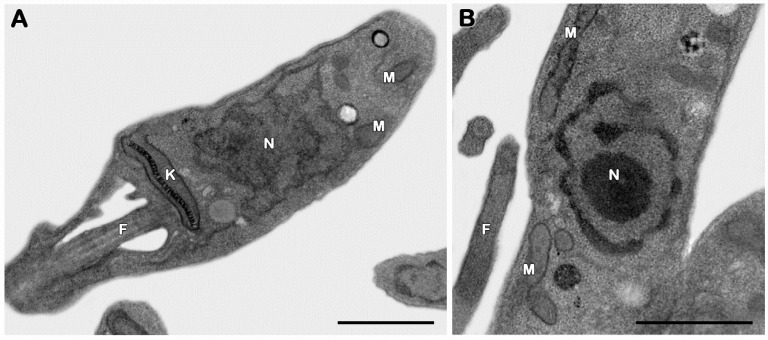
Ultrastructural analysis of *Leishmania* (*L.*) *amazonensis* promastigotes. (**A**,**B**) Untreated parasites showing typical elongated morphology with the normal kinetoplast (K), mitochondrion (M), flagella (F), and nucleus (N). Bars = 1 μm.

**Figure 6 tropicalmed-08-00324-f006:**
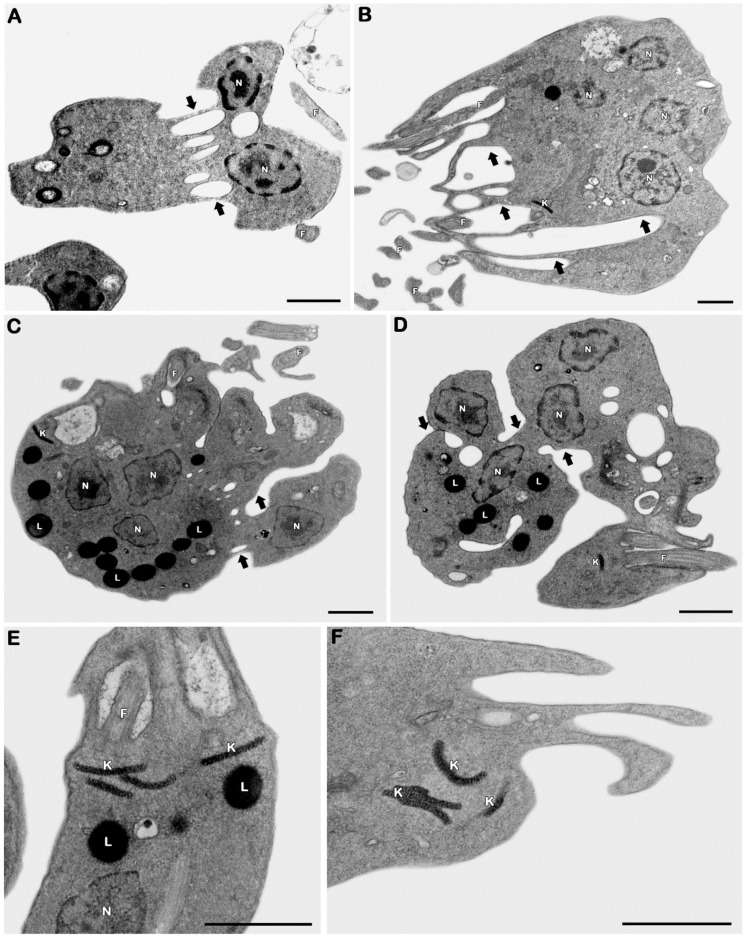
Ultrastructural effects of 3CR in *Leishmania* (*L.*) *amazonensis* promastigotes. (**A**–**F**) The treatment with 488.1 mM promoted the appearance of parasites with (**A**–**D**) multiple nuclei (N) and (**E**,**F**) kinetoplasts (K) parasites, also showing a high number of lipid droplets (L) (**C**–**E**). Treated promastigotes also presented cytosolic elongated invaginations (black arrows). F: flagella. Bars = 1 μm.

**Table 1 tropicalmed-08-00324-t001:** Mean hydrodynamic diameter (HD) and polydispersion index (PDI) of the micelles, determined by dynamic light scattering (DLS).

Formulations	T ^1^ (°C)	HD (nm)	PDI	T (°C)	HD (nm)	PDI
P407	25	24.16 ± 0.42	0.314 ± 0.014	37	18.48 ± 0.15	0.206 ± 0.011
P407-3CR	25	26.81 ± 0.82 *	0.463 ± 0.023	37	21.17 ± 0.45 *	0.326 ± 0.027

^1^ T: Temperature.Data represent the mean of three independent experiments ± standard error (SE). * Statistical difference between the micellar system with and without 3CR (*p* < 0.05).

**Table 2 tropicalmed-08-00324-t002:** In vitro activity of 3CR and micelles in *Leishmania* (*L.*) *amazonensis* promastigotes after 48 h of treatment.

Formulations	IC_50_/48 h (mM) ^2^
3CR ^1^	488.1 ± 3.7 ^3^
P407	-
P407-3CR	419.9 ± 1.5
Amphotericin B	0.0002 ± 0.1

^1^ 3CR: 3-Carene, ^2^ IC_50_/48 h: half of the maximum inhibitory effect at 48 h of treatment. ^3^ Data are expressed as mean ± standard error (*n* = 3).

**Table 3 tropicalmed-08-00324-t003:** In vitro activity of 3CR and micelles in *Leishmania* (*L.*) *amazonensis* intracellular amastigotes and primary macrophages after 72 h of treatment.

Formulations	IC_50_/72 h ^2^ (mM)	CC_50_/72 h ^3^ (mM)	SI ^4^
3CR ^1^	>1.5 **^5^**	5.2 ± 0.5	-
P407	1.4 ± 0.1	2.9 ± 0.2	2.0
P407-3CR	0.7 ± 0.1	3.7 ± 0.1	5.2
Amphotericin B	0.0003 ± 0.0	>0.02	>66.6

^1^ 3CR: 3-Carene. ^2^ IC_50_ /72 h: half of the maximum inhibitory effect at 72 h of treatment. ^3^ CC_50_ /72 h: half of the maximum cytotoxic effect at 72 h of treatment. ^4^ SI: Selectivity index determined by equation CC_50_/IC_50_ at the same time of treatment. ^5^ Data are expressed as mean ± standard error.

## Data Availability

All data generated or analyzed during this study are included in this published article.

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
