# Peer review of "Effect of 3-Carene and the Micellar Formulation on Leishmania (Leishmania) amazonensis"

_tropicalmed, 2023, doi:10.3390/tropicalmed8060324_

Round 1
Reviewer 1 Report
1) Consider a shorter title to express the main contribution of the article.
2) IC50 values should be expressed in M unit since is it a defined substance (3CR - 3,7,7-Trimethyl-bicyclo [4,1,0]hept-3-ene, 96.1% pure). This change is important to allow the comparison of potency among different compounds. The unit of g/L should be used only for extracts.
3) Some of the cited articles are not the most appropriate ones to be used as a reference for the content as it refers. Please, revise the cited articles for more specialized literature related to the content of the statement. Ie: citation nº 3 – Review of general aspects about Leishmania as used as a reference for a Leishmania tissue tropism.
4) Revise the use of units in the text and uniformized the style. Ie: the same antibiotic is written in different manners. Line 125: 100 µg/mL of streptomycin Line 128: 5 mg streptomycin/mL.
5) The most common assay to evaluate amastigote uses THP-1 cells. The authors could explain the choice of intraperitoneal macrophages.
6) The protocol number of the ethical committee of animal use was not included.
7) The number of 100 macrophages per condition should not be considered enough to demonstrate with reasonably certain the result.
Please consider performing the experiment using 100 cells per well and at least 3 wells per condition (technical triplicates) for each biological replicate, since quality controls like Z' factor are not provided.
About HTS F’factor, see: https://doi.org/10.1177/108705719900400206
8) The assay with macrophages was performed in 72h. Therefore, cytotoxic assay for all cell lines should apply the same time of incubation to be considered valid.
9) Beyond the value of IC50 itself, the dose-response curves are very meaningful and informative.
Please, include the graphs following ‘Guidelines for EC50/IC50 estimation’ (doi: 10.1002/pst.426) for instance or other relevant references to do so.
The replicates should be demonstrated by overlapping the data in the graph.
10) Risarzurin methods evaluate metabolism, which is not absolute proof of death. Thus, the word leishmanicidal should not apply in this context. For more information about static/cidal assays, see: 10.1371/journal.pntd.0004584
11) To progress in the drug discovery pipeline, a compound must have at least an SI of 10.
In line 264 and 265, the authors conclude the P407-3CR is not toxic, however, de SI show otherwise.
12) The article includes a very good discussion about the possible mode of action of 3CR regarding other treatments.
Consider including microscopy images of treatments with different modes of action, to demonstrate how 3CR is different/similar from others. Confocal immunofluorescence should be enough to demonstrate these differences. This additional experiment will reinforce the discussion.
Revise the text more accurately regarding English and scientific writing.
Author Response
1) Consider a shorter title to express the main contribution of the article.
Answer: Following reviewer recommendation, the new title is “3-Carene Impairs Cytokinesis and the micellar formulation improves its Leishmanicidal activity in Leishmania (Leishmania) amazonensis”.
2) IC50 values should be expressed in M unit since is it a defined substance (3CR - 3,7,7-Trimethyl-bicyclo [4,1,0]hept-3-ene, 96.1% pure). This change is important to allow the comparison of potency among different compounds. The unit of g/L should be used only for extracts.
Answer: The IC50 values were converted to M as recommended.
3) Some of the cited articles are not the most appropriate ones to be used as a reference for the content as it refers. Please, revise the cited articles for more specialized literature related to the content of the statement. Ie: citation nº 3 – Review of general aspects about Leishmania as used as a reference for a Leishmania tissue tropism.
Answer: The references and citations of the introduction were carefully revised, and some references were replaced.
4) Revise the use of units in the text and uniformized the style. Ie: the same antibiotic is written in different manners. Line 125: 100 µg/mL of streptomycin Line 128: 5 mg streptomycin/mL.
Answer: all units were corrected.
5) The most common assay to evaluate amastigote uses THP-1 cells. The authors could explain the choice of intraperitoneal macrophages.
Answer: Actually, both THP-1 and peritoneal macrophages are extensively employed for these assays. And both models present advantages and limitations. Our group opts to primary cells in order to lineages due to the proximity to real state of host cell. Additionally, THP-1 is a monocyte lineage that must be differentiated into macrophages, one additional step in relation to peritoneal cells.
6) The protocol number of the ethical committee of animal use was not included.
Answer: The section 2.11 “Ethics statement” was included in methods.
7) The number of 100 macrophages per condition should not be considered enough to demonstrate with reasonably certain the result. Please consider performing the experiment using 100 cells per well and at least 3 wells per condition (technical triplicates) for each biological replicate, since quality controls like Z' factor are not provided.
About HTS F’factor, see: https://doi.org/10.1177/108705719900400206
Answer: We´re sorry for the text not clear in methods. We performed all three biological experiments with 2 technical replicates in each one. In our lab, we compared the quantification in total of 100, 200, 300 and 400 cells per experimental condition, and no statistical differences were observed (personal communication). The corrected sentences were included in the methods section.
8) The assay with macrophages was performed in 72h. Therefore, cytotoxic assay for all cell lines should apply the same time of incubation to be considered valid.
Answer: We totally agree with the reviewer. In accordance with this point, the SI values were calculated comparing IC50 and CC50 values for macrophages in 72h. As it was answered in point #5, the use of primary cells was chosen as our gold standard method. In this context, L929 cells were used as additional information about toxicity, confirming macrophages data.
9) Beyond the value of IC50 itself, the dose-response curves are very meaningful and informative.Please, include the graphs following ‘Guidelines for EC50/IC50 estimation’ (doi: 10.1002/pst.426) for instance or other relevant references to do so. The replicates should be demonstrated by overlapping the data in the graph.
Answer: The IC50 graphs were included, and figures were created.
10) Risarzurin methods evaluate metabolism, which is not absolute proof of death. Thus, the word leishmanicidal should not apply in this context. For more information about static/cidal assays, see: 10.1371/journal.pntd.0004584
Answer: There are different methods for analysis parasites viability for sure. Our group has been employing rezasurin method for leishmanicidal assays at least in the last 10 years. For sure, the term “leishmanicidal” is not the best one once promastigotes proliferate, we agree. Due to this point we carefully used the term. Example: in abstract, we said “This study aimed to develop Poloxamer 407 micelles capable of delivering 3CR (P407-3CR) to improve antileishmanial activity” generally, but we also said “3CR and P407-3CR inhibited the growth of L. (L.) amazonensis promastigote with IC50/48h of 488.1±3.7 and 419.9 ± 1.5 mM, respectively”, when we specified this results obtained by this assay. All citations of this result was carefully revised in the whole text. Anyway, the use of rezasurin for this purpose has been well-documented in the scientific community (Salazar-Villamizar ME, Escobar P, 2022. Exp Parasitol. doi: 10.1016/j.exppara.2021.108206; Sirak B et al, 2021. Molecules, 10;26(24):7473. doi: 10.3390/molecules26247473; Tadele M et al 2021. BMC Pharmacol Toxicol 22(1):71; doi: 10.1186/s40360-021-00538-2; Koutsoni OS et al 2019. Bio Protoc 9(21):e3410. doi: 10.21769/BioProtoc.3410; among many others).
11) To progress in the drug discovery pipeline, a compound must have at least an SI of 10.
In line 264 and 265, the authors conclude the P407-3CR is not toxic, however, de SI show otherwise.
Answer: Toxicity is totally dependent of the comparison employed; some studies and technical notes describe SI>50 for the following steps in the pipeline. We think this stringency is very necessary in order to find a real promising prototype for clinical intervention. In lines 264 and 265, the data about toxicity in L929 cells were described, is not referred to SI in macrophages. Once we totally agree with the reviewer, the sentence “Anyway, the SI values for 3CR in micelles or not were far from ideal, much lower than SI for amphotericin B.” was included in the cytotoxicity paragraph of the discussion.
12) The article includes a very good discussion about the possible mode of action of 3CR regarding other treatments.
Consider including microscopy images of treatments with different modes of action, to demonstrate how 3CR is different/similar from others. Confocal immunofluorescence should be enough to demonstrate these differences. This additional experiment will reinforce the discussion.
Answer: Once again, we agree with the reviewer. Actually, we tried to. However, our primary antibody did not work unfortunately. Now, the batch of 3CR is over, and the first author left the lab to a post doc in another place. We are thinking about it, and we hope we can perform some experiments of validation in a near future.
Reviewer 2 Report
The manuscript is well designed and well written. It needs a minor revision specially the discussion section the authors repeated an introductory part about the study and I suggest to remove the lines from 505 to line 329. I have an question to the authors why the authors cited their published work about the same parasite (https://www.sciencedirect.com/science/article/pii/S1773224723002289). So, the authors should be differentaite between the two papers.
Author Response
The manuscript is well designed and well written. It needs a minor revision specially the discussion section the authors repeated an introductory part about the study and I suggest to remove the lines from 505 to line 329. I have an question to the authors why the authors cited their published work about the same parasite (https://www.sciencedirect.com/science/article/pii/S1773224723002289).So, the authors should be differentaite between the two papers.
Answer: The first 2 paragraphs of the discussion section were strongly reduced, following the recommendation of the reviewer. We only maintained few sentences to introduce the discussion points. The previous article deals with a pre-formulation study using 3-carene to identify whether combinations and which combinations of poloxamers could be more efficient to deliver 3-carene. In the present work, we used the selected poloxamer P407 to formulate micelles and improve the anti-leishmanial activity of 3-carene. We then included more detailed physicochemical and biological evaluations to investigate the compound and formulations. In this sense, we evaluated the particle size and rheological behavior, which can influence activity, and different biological parameters were evaluated such as toxicity in more than one type of mammalian cell and leihsmanicidal effect on intracellular amastigotes and ultrastructural investigation of the mechanism of action of 3-carene in the parasite.
Reviewer 3 Report
The manuscript is well written and reports interesting data about the leishmanicidal activity of P407-3CR. I sugestão that this article be accepted.
Additional comments:
The manuscript reports the leishmanicidal activity of 3-Carene (3CR) and Poloxamer 407 micelles capable of delivering 3CR (P407-3CR). I consider the topic relevant for the development of new therapeutic alternatives for leishmaniasis, a neglected disease that has a very limited therapeutic arsenal. The manuscript reports good results about the leishmanicidal effect of P407-3CR, showing that 3-carene induces a bizarre multiple nuclei and kinetoplast phenotype in Leishmania amazonensis. The manuscript shows the ability of the use of Poloxamer 407 micelles to increase the leishmanicidal pharmacological activity of 3CR, in addition to describing the ultrastructural alterations that 3CR promotes in the parasite. The authors can add the protocol number of the ethical committee of animal use and revise to correct some units of measurement. I consider that the controls used by the authors are sufficient. The conclusions are consistent with the evidence and arguments presented in the manuscript. I consider that the references are appropriate. I consider the figures and tables in the manuscript to be appropriate.
Author Response
Additional comments:
The manuscript reports the leishmanicidal activity of 3-Carene (3CR) and Poloxamer 407 micelles capable of delivering 3CR (P407-3CR). I consider the topic relevant for the development of new therapeutic alternatives for leishmaniasis, a neglected disease that has a very limited therapeutic arsenal. The manuscript reports good results about the leishmanicidal effect of P407-3CR, showing that 3-carene induces a bizarre multiple nuclei and kinetoplast phenotype in Leishmania amazonensis. The manuscript shows the ability of the use of Poloxamer 407 micelles to increase the leishmanicidal pharmacological activity of 3CR, in addition to describing the ultrastructural alterations that 3CR promotes in the parasite. The authors can add the protocol number of the ethical committee of animal use and revise to correct some units of measurement. I consider that the controls used by the authors are sufficient. The conclusions are consistent with the evidence and arguments presented in the manuscript. I consider that the references are appropriate. I consider the figures and tables in the manuscript to be appropriate.
Answer: The section 2.11 “Ethics statement” was included in methods.
Reviewer 4 Report
This is an interesting manuscript, that is nicely written with good references. The data are interesting an nicely presented; it was a pleasure to read this. I do have a few minor questions/recommendations as indicated below:
section 2.2:; are the micelle preparations filter sterilized before use?
line 152: Is there a reference for the panoptic stain and who is the supplier?
line 161 is not a sentence
line 180 : treated with what?
line 234: are the values (indicated as similar activity) statistically the same or different? please clarify here and in Table 2
line 284: change 'of' to 'on'
line 324: change 'is' to 'has been'
line 371: the meaning of '...with seen more..' is unclear
line 434: should this be 'cell' or 'cells' arrest?
line 477: date for this citation?
nicely done
Author Response
This is an interesting manuscript, that is nicely written with good references. The data are interesting an nicely presented; it was a pleasure to read this. I do have a few minor questions/recommendations as indicated below:
section 2.2:; are the micelle preparations filter sterilized before use?
line 152: Is there a reference for the panoptic stain and who is the supplier?
line 161 is not a sentence
line 180 : treated with what?
line 234: are the values (indicated as similar activity) statistically the same or different? please clarify here and in Table 2
line 284: change 'of' to 'on'
line 324: change 'is' to 'has been'
line 371: the meaning of '...with seen more..' is unclear
line 434: should this be 'cell' or 'cells' arrest?
line 477: date for this citation?
Answer: all minor points raised by the reviewer were corrected. In relation to preparation of micelles, sterile PBS was used, but it was not filtered.
Reviewer 5 Report
The study is well designed and organized, and the manuscript is mostly well written; however, the following minor points should be better to be considered.
Comments:
- In the text (Abstract, Keywords, Introduction, Discussion), the mixed usage of the terms, “antileishmania” and “antileishmanial” is described at different parts/lines; “antileishmanial” would be recommended.
-Against the scientific name of the genus Leishmania, better to be provided the subgenus names in italic, e.g., Leishmania (Leishmania) amazonensis at first then, L. (L.) amazonensis; L. (L.) donovani, L. (L.) infantum, L. (L.) major; L. (Viannia) braziliensis, etc.
- Line 38, replace Leishmaniasis by Leishmaniases
- Line 39; after Leishmania, insert “and transmitted by blood-sucking phlebotomine sand flies as vector-borne disease”.
- In the text (Abstract, Keywords, Introduction, Discussion), the mixed usage of the terms, “antileishmania” and “antileishmanial” is described at different parts/lines; “antileishmanial” would be recommended.
-Against the scientific name of the genus Leishmania, better to be provided the subgenus names in italic, e.g., Leishmania (Leishmania) amazonensis at first then, L. (L.) amazonensis; L. (L.) donovani, L. (L.) infantum, L. (L.) major; L. (Viannia) braziliensis, etc.
- Line 38, replace Leishmaniasis by Leishmaniases
- Line 39; after Leishmania, insert “and transmitted by blood-sucking phlebotomine sand flies as vector-borne disease”.
Author Response
The study is well designed and organized, and the manuscript is mostly well written; however, the following minor points should be better to be considered.
Comments:
- In the text (Abstract, Keywords, Introduction, Discussion), the mixed usage of the terms, “antileishmania” and “antileishmanial” is described at different parts/lines; “antileishmanial” would be recommended.
-Against the scientific name of the genus Leishmania, better to be provided the subgenus names in italic, e.g., Leishmania (Leishmania) amazonensis at first then, L. (L.) amazonensis; L. (L.) donovani, L. (L.) infantum, L. (L.) major; L. (Viannia) braziliensis, etc.
- Line 38, replace Leishmaniasis by Leishmaniases
- Line 39; after Leishmania, insert “and transmitted by blood-sucking phlebotomine sand flies as vector-borne disease”.
- In the text (Abstract, Keywords, Introduction, Discussion), the mixed usage of the terms, “antileishmania” and “antileishmanial” is described at different parts/lines; “antileishmanial” would be recommended.
-Against the scientific name of the genus Leishmania, better to be provided the subgenus names in italic, e.g., Leishmania (Leishmania) amazonensis at first then, L. (L.) amazonensis; L. (L.) donovani, L. (L.) infantum, L. (L.) major; L. (Viannia) braziliensis, etc.
- Line 38, replace Leishmaniasis by Leishmaniases
- Line 39; after Leishmania, insert “and transmitted by blood-sucking phlebotomine sand flies as vector-borne disease”.
Answer: all minor points raised by the reviewer were corrected.
Round 2
Reviewer 1 Report
Dear authors,
I appreciate the considerations and changes that were performed. An improvement in compound formulations is needed in drug discovery for NTD. We are losing several molecules due to poor physicochemical properties and studies like yours reinforce the effort in this sense. However, the data about how significant the improvement of the use of micelles on antileishmanial 3CR is not strong.
Figure 3C, demonstrated clearly that P407 itself have a similar effect on L. amazonesis that P407-3CR. The IC50 of 1.4 ± 0.1 mM (P407 alone) in comparison to 0.7 ± 0.1 mM of P407-3CR is very similar to claim in the title “micellar formulation improves its Leishmanicidal activity”. Also, the curve was not well-fitted enough to ensure the data.
However, that came to my attention that the dispersion on micelles is not always uniform, and for that units in M not always could be applied to nanostructure studies. Could the author provide a table regarding an equivalent unit of dose of the compound? (For ie: doi: 10.1038/srep13500). This kind of biological correlation allows the reader properly compare the potency of the compound in comparison to established drugs.
Table 2 and Table 3: All units should be shown at mM in order to not confuse the reader.
In Figure 2 the authors describe in the legend 'the dashed line represents 50 % inhibition of proliferation'. However, there are no dashed lines in the graph.
Please, adjust the graph size (graph B is smaller than the others) in Figure 2 and Figure 3.
Figure 3 legend: Considerer rewrite the legend to become less repetitive (in terms of names of the compounds) and more informative about the result itself ie: Antiamastigote activity is improved by…
Carefully revise the text, since some of the text changes could lead to minor errors such as space suppression or extra letters in the words ie: ‘antileishmaniall’
'Balb/c' word must be changed for the correct term 'BALB/c'
About the comment about the use of THP-1 cells, in respect to the principles of the 3Rs (Replacement, Reduction and Refinement) in animal research it is strongly discouraged the use of primary cells when other methods could provide similar results. I respect your method decision, although it should be considered to comply with the Replacement principle for future studies.
Author Response
All the minor points raised by the reviewer were corrected as suggested. In relation to the improvement of the antileishmanial activity by the micelle, we agree with the reviewer, and changed the title to “The effects of 3-Carene and the micellar formulation on Leishmania (Leishmania) amazonensis”. In relation to the dispersion of micelles, to determine the exact content of 3CR in the micelles, it would be necessary to have carried out the encapsulation efficiency test. However, it was not possible to carry out this test, one of the limitations of our study. Then, we assume that the whole added 3CR has been incorporated into the micelles, resulting in a 100% loading.